# Protocol Development of a Personalized Balanced Nutrition Concept for Preschool Children, Primarily Those with Food Allergies, Using an IT Platform

**DOI:** 10.3390/medicina59081367

**Published:** 2023-07-26

**Authors:** Siniša Košćina, Adrijana Miletić Gospić, Ivana Banić, Domagoj Sabljak, Marcel Lipej, Tamara Birkić, Davor Plavec, Tomislav Marjanović, Darja Sokolić, Mirjana Turkalj

**Affiliations:** 1Healthcare and Public Sector, IN2 Group, HR-10000 Zagreb, Croatia; 2Department of Nutrition, Srebrnjak Children’s Hospital, HR-10000 Zagreb, Croatia; adrijana_miletic@yahoo.com; 3Department for Translational Medicine, Srebrnjak Children’s Hospital, HR-10000 Zagreb, Croatia; 4Research Department, Srebrnjak Children’s Hospital, HR-10000 Zagreb, Croatia; 5IT Department, Srebrnjak Children’s Hospital, HR-10000 Zagreb, Croatia; mlipej@bolnica-srebrnjak.hr; 6Department of Pediatrics, Faculty of Medicine, J.J. Strossmayer University of Osijek, HR-31000 Osijek, Croatiamturkalj@bolnica-srebrnjak.hr (M.T.); 7Croatian Agency for Agriculture and Food, HR-31000 Osijek, Croatia; darja.sokolic@hapih.hr; 8School of Medicine, Catholic University of Croatia, HR-10000 Zagreb, Croatia; 9Department for Allergology and Pulmonology, Srebrnjak Children’s Hospital, HR-10000 Zagreb, Croatia

**Keywords:** immunology, allergy, food allergy, nutrition, allergy registry

## Abstract

Children with food allergies are at higher risk for severe anaphylactic reactions and for key nutrient deficiency. In order to address these concerns, enable early detection, and improve the monitoring of children with food allergies, an innovative IT platform will be developed by IT experts (IN2 Ltd. Zagreb, Croatia, part of Constellation Software Inc. (Toronto, ON, Canada)) and Srebrnjak Children’s Hospital, Zagreb, Croatia (SCH) for the effective implementation of personalized balanced nutrition in preschool institutions in Croatia. Additionally, the data obtained through this research, including epidemiological data on allergic diseases, clinical data (diagnostic allergy tests and others), anthropometry, and physical activity status, will be used to create a national Allergy registry. Other than being a tool for personalized and balanced nutrition for children, especially those with special dietary requirements (including food allergy and intolerance), the IT platform developed in this study will enable the continuous monitoring of these children as a part of their clinical management plan and earlier detection of food allergies, intolerance, and other conditions, even outside of the healthcare system. This research also aims at optimizing current and developing novel personalized therapeutic regimes, detecting novel early biomarkers in children with food allergies and intolerances, and involving all key stakeholders (caregivers, preschool institutions, etc.) in the shared-care approach in the management of food allergies in children.

## 1. Introduction

In the last few decades, there has been a growing worldwide trend of eating disorders and conditions with special dietary requirements in the pediatric population, such as obesity, food intolerance, and food allergies [1]. This emphasizes the need to implement healthy eating habits for children from an early age, a key prerequisite for healthy growth and development. A body of evidence indicates that nutrition is one of the major factors influencing the increase in the prevalence of various diseases, such as allergies, obesity, diabetes, and metabolic and cardiovascular diseases [2,3,4]. Starting from the prenatal period through early life and childhood to adulthood, food, among other things, affects the composition of the gastrointestinal (GI) microbiome, a significant part of the body’s immune system. The GI microbiome is an ecosystem in which the interaction between humans and GI bacteria defines the balance of immune responses manifested in tolerance to harmless food allergens [5,6,7]. Food allergy is a public health issue that affects mostly children, and it has been increasing in prevalence in the last 30 years [8,9]. Furthermore, the rate of food allergy is constantly increasing, especially in the pediatric population. In Western countries, it affects up to 8% of children [10]. The symptoms can vary from mild to life-threatening allergic reactions. Currently, the management of food allergy involves allergen avoidance and emergency (rescue) treatment. The eight most common food allergens in general but also in the Republic of Croatia are milk, eggs, peanuts, soy, wheat, tree nuts, fish, and shellfish, all of which are frequently consumed in Croatia [11]. Children, their families as well as preschool institution staff must remain constantly alert, which greatly affects their quality of life. Most preschool and school staff members are not educated or trained in managing food allergy reactions, especially in the application of an epinephrine auto-injector in case of a life-threatening reaction [12]. On the other hand, the introduction of an exclusion diet represents a risk for nutritional deficits, which is far more pronounced in children and their special growth-related requirements. Moreover, non-allergic food reactions, such as food intolerance, are commonly mistaken for food allergies, and the introduction of an unnecessarily strict diet in such children should also be avoided [13,14]. Additionally, modern allergy diagnostic tests such as component-resolved diagnostics (CRD) allow for better identification of allergens enabling a personalized approach in nutrition counseling. E.g., certain patients allergic to fruit should not be on a strict elimination diet to specific fruit, or rather they should eliminate only fresh, but not thermally processed fruit [7].

The main objective of this research is to develop, adapt and implement an information system that will enable a user-friendly application of completely customized and balanced menus for preschool children in daycare facilities following their individual needs, as well as to adopt healthy eating habits keeping in mind that children in Croatia eat most of their daily meals (on work days) in preschool facilities which have to ensure 80% of daily energy requirements for children in full-time daycare. Moreover, infancy and early childhood is the key time for introducing and adopting proper dietary habits, thus potentially preventing the development of common chronic diseases in the future [15].

Such customized and balanced menus are especially important for children with food allergies and intolerance who are at high risk of malnutrition and impaired growth and development. However, children with food allergies are able to achieve similar mean nutrient intakes as healthy (non-allergic) children if they have previously received proper nutrition counseling and are provided with a substitution of nutritionally equivalent foods [7].

This system (the IT platform) integrates the food and allergen database, allowing for the creation of standard age-appropriate menus with adequate energy and nutritive values, automatically adjusting for specific food allergens. These children will be promptly provided with balanced alternative menus designed by a clinical team at SCH (physicians- allergy specialists, and nutritionists). The system will, among other things, establish a communication link between the Srebrnjak Children’s Hospital (SCH), the National Reference Center for Clinical Allergy in Children of the Croatian Ministry of Health, and preschool institutions, which will enable the exchange of data in real-time and allow for the possibility of timely adjustment of the menu. A schematic representation of the key stakeholders and workflow in the IT platform is shown in Figure 1.

Additionally, based on the collected data (clinical and other), a national Allergy registry will be developed, which will enable the implementation of optimal disease management, the improvement of healthcare and other non-related services and policies (such as the food industry) in the form of individually or personally tailored nutrition, with a special focus on allergenic content inside each meal and its components.

## 2. Materials and Methods

Research and development activities within this study will take place in preschool institutions in the Republic of Croatia and SCH over 33 months and encompass six different phases: (1) development of the information system and data integration, (2) recruitment and classification of target groups of preschool children according to their allergy status, (3) anthropometric monitoring of the development of preschool children, (4) establishment of an electronic Allergy register, (5) implementation of an IT platform in preschool institutions, and (6) assessment of the platform validity. 

The implementation of the project will begin with the design and development of an information system for the integration of nutritional data, which will be built by the project partner IN2 Ltd. 

### 2.1. Study Population

A minimum of 450 pediatric participants will be recruited in a prospective, non-interventional type of clinical study by SCH. Participants will be recruited in preschool institutions in 3 distinct geographical regions in Croatia (a minimum of 150 participants in each region), differing in a number of environmental and lifestyle factors, including dietary habits. The recruitment will take place during the academic year, which is from the beginning of September until mid-June in Croatia. Informed written consent will be obtained from the children’s parents/caregivers. The study protocol was approved by the local Ethics Committee (at SCH). 

The study will include participants (children aged 1 to 7 years- typical age for preschool children in Croatia, of both sexes, regardless of their allergy background) whose parents/caregivers signed an informed consent, i.e., agreed to participate in the study after receiving both written and oral information about the study to their satisfaction and do not meet the criteria for exclusion. Criteria for exclusion will be assessed by a specialist physician (pediatric allergy specialist). The children will be enrolled in the study after their parents/caregivers have read, agreed to, and signed the informed consent form. The subjects will be assessed for inclusion and exclusion criteria on the first study visit after signing the informed consent. Participants involved in the study will be able to use their regular therapy, i.e., conduct treatment and all other non-therapeutic measures, as if they were not involved in the study (real-life study).

Exclusion criteria are known congenital and other serious chronic illnesses (such as malignant diseases, chromosomal aberrations, neurological disorders that affect oral motor skills, chronic gastrointestinal disorders, or severe metabolic disorders). Moreover, children will be excluded from study visits and biomaterial collection in the case of a fever of at least 38.5 °C during the last two weeks prior to the planned visit.

### 2.2. Assessments, Measurements, Diagnostic Procedures and Data Collection

Upon recruitment, after their parents/caregivers have agreed to participate in the study and signed the informed consent form, participants will undergo anthropometric measurements to evaluate their nutritional status. This will include the measurement of height and weight for the calculation of BMI (body mass index) and BMI percentiles, as well as two-site skinfold measurements (triceps and subscapular) to assess body composition. All participants will undergo a skin prick test (SPT) on a standard palette of food and inhaled allergens (listed in Appendix A). Participants whose parents/caregivers have agreed to it will undergo blood sampling for total IgE detection. 

A subset of participants with positive SPT and risk for food allergy/intolerance according to personal and family medical history will be invited to additional visits and diagnostic procedures at SCH. These participants will undergo a physician examination (pediatric allergy specialist) and a battery of diagnostic tests and procedures indicated by the physician to establish a diagnosis of food allergy. These will include SPT to additional allergens (if needed), total and allergen-specific immunoglobulin E (sIgE, basophil activation test (BAT), component-resolved diagnostics (CRD) and/or ImmunoCAP Immuno-Solid phase Allergy Chip (ISAC) tests, oral food challenge tests (OFC)- (sIgE, BAT, CRD and ISAC as indicated by an allergy specialist, lung function tests and FENO (if needed), blood count and biochemistry assays, atopy patch test (APT), oral food challenge tests (OFC), etc. In order to identify additional conditions that might affect and aggravate the underlying disease, a proportion of participants (when indicated) will be tested for common food allergy comorbidities, such as gastroesophageal reflux disease (GERD), obstructive sleep apnea syndrome (OSAS) as well as asthma, allergic rhinitis, and atopic dermatitis, including anamnesis data taken from the child’s parents/caregivers.

Peripheral whole blood samples will be collected by venipuncture into EDTA-coated vacutainers (for hematology analyses) and into vacutainers with clot activator and gel for serum separation (for biochemistry and certain allergy assays). During this study, a total of 10.5 mL peripheral blood samples per participant maximum will be collected at the recruitment visit and/or at additional visits at SCH. The remainder of blood samples (in EDTA-coated vacutainers) and sera left over after diagnostic tests will be stored at −20 °C for subsequent analyses (ISAC, nutritional status).

### 2.3. Follow Up

If necessary, participants with food allergies will be followed up as a part of their clinical management plan on average every 6 months or even more often if they had severe acute episodes. Follow-up visits will involve diagnostic procedures, as indicated: allergy testing (skin prick test-SPT, total and allergen-specific immunoglobulin E-IgE, lung function testing, FENO, basophil activation test-BAT, blood count and biochemistry assays, oral food challenge test). These tests and procedures will be carried out at regular intervals (usually once a year) and if needed (if indicated by a pediatric allergy specialist). Additionally, an assessment of the participant’s nutritional status and other tests (such as vitamin status-vitamin D3, vitamins A, and E) will be performed on children at risk, their dietary habits will be reviewed (by food diaries), and they will undergo counseling by a clinical nutritionist if needed.

### 2.4. Questionnaires

In order to screen the preschool population in Croatia for food allergy and intolerance and assess the risk for food allergy development, several questionnaires will be used and adapted based on existing validates ones (ISAAC and questionnaires used in the FP7 Europrevall-iFAAM project) [16,17].

### 2.5. Statistical Analysis

Features containing string notations will be numerically encoded. For selected individuals (those at risk), data will be divided into time periods, i.e., baseline and follow-up data. Certain features, such as those describing allergic sensitization, will be converted to binary or ordinal features (e.g., elevated/normal, low/normal/high, yes/no, etc.). Missing data above a certain threshold for a single variable (>10%) will be excluded from the analysis. 

Shapiro-Wilk’s normality test will be used to test the distribution of data for normality. Data with normal distribution will be expressed by arithmetic mean and standard deviation (mean ± SD), and those with asymmetric distribution by median and range. According to the normality of the data distribution for each variable (set), parametric or non-parametric statistical tests will be used. Statistical analysis will be performed using the STATISTICA 12 program (StatSoft, Tulsa, OK, USA). Values of *p* < 0.05 will be considered significant.

## 3. Results

The data that will be collected in this study are shown in Table 1.

Based on the completed standardized questionnaires and performed SPT, target groups of preschool children will be stratified according to their allergy status: (a) children with an established diagnosis of food allergy; (b) children at risk of developing mild, moderate, or severe allergic reactions based on atopic background; (c) children with no diagnosis of food allergy or intolerance; and (d) children with a likely diagnosis of food allergy/intolerance based on symptoms [18]. The latter group will be invited for further allergy specialist evaluation, diagnostics, and nutritionist evaluation and counseling at SCH (as described in the Assessments, measurements, diagnostic procedures, and data collection section). Additionally, we will anthropometrically monitor all groups of participants, monitor their physical activity level with smart bands, and examine their nutritional status (vitamin D3, A, and E detection and using validated questionnaires) to ensure that a balanced or specifically modified diet in target groups A, B, and D does not adversely affect their growth and development and that it meets their nutritional needs.

In doing so, care will be taken to differentiate between children with food allergies from children with intolerance to specific food. This will contribute to a more precise diagnosis of these conditions, which is problematic even among pediatric allergy specialists, despite the fact that proper management and provision of adequate nutrition are of utmost importance in both food allergy and food intolerance [19].

Data collected during this study will be integrated into the information system for monitoring the child’s development over time, with availability to all stakeholders, each within their own role, application module, and process step. In this way, parents/caregivers will get an insight into their child’s measurements, physical activity, and specific dietary requirements immediately, and thus the needs of their child. In case of an observed problem with the child’s growth (i.e., deviation from growth standards and references) and development or insufficient physical activity, the system will allow timely response in the context of modifying the child’s dietary habits and menus at preschool institutions and provide guidelines related to improving nutrition and physical activity designed by healthcare professionals at SCH. The platform will allow for continuous data collection and access to adequate stakeholders (parents/caregivers, healthcare providers, preschool institutions), including new-onset food allergy/intolerance, possible delays in establishing and confirming a diagnosis of food allergy, and the development of clinical tolerance to specific allergens, in order to timely adjust the child’s nutrition.

## 4. Discussion

Due to the rising prevalence of food allergy and intolerance, it is obvious that children’s eating habits need to be adjusted to allergies and other disorders according to professional standards and guidelines. A significant problem in educational and daycare institutions is the lack of staff who are trained and qualified to take care of children with food allergies and food intolerance and to compile and adjust their diet. Today, parents of children with food allergies are most often required to bring food from home that is safe for their child to consume. This, in turn, can result in a uniformly unbalanced menu, stigmatization by peers, and unnecessary financial costs for the child’s family. From the perspective of preschool institutions, there is a significant but unfulfilled demand for know-how associated with both healthy and specially adapted diets.

This IT platform aims to provide balanced age-appropriate menus to all children in preschool institutions, especially children with food allergy as, a particularly vulnerable population at high risk of malnutrition and growth impairment and with impaired quality of life, as this is essential in the management of children with food allergy [7]. Hence, this study will address the issues that significantly impact the quality of life of children with food allergy and food intolerance and their families, as well as potentially cause growth and development impairment for these children. Moreover, we aim to establish novel and improved prevention strategies and risk mitigation strategies for the development of nutritional deficiencies and severe allergic reactions in allergic children due to accidental exposure or cross-contamination. This will be performed with the support of a web service-based innovative information system, multiplatform applications, and Allergy registry database creation, which will optimize the collection, processing, and availability of data, as well as the processes involved in management (treatment) and prevention issues resolving. 

Furthermore, the development of an allergy registry that provides epidemiologic data on allergic reactions in the general population will help revise current and develop novel and improved guidelines, strategies, and policies on allergy prevention, build awareness and knowledge on the issue in the general public and enable the successful transfer of relevant knowledge for the clinical practice to educational and daycare institutions. Additionally, this database will create added value in the context of clinical and translational research, ultimately contributing to the development of personalized and individually tailored management and prevention plans. The mechanisms that lead to allergic reactions to food in certain people, the vast heterogeneity in symptoms among allergy sufferers, and factors that potentiate the development of food allergies in early childhood are still largely unclear, and establishing such a repository (allergy database) is a key step towards personalized treatment of conditions requiring special nutritional needs.

Within this study, tight and effective cooperation between healthcare providers, scientific researchers, preschool institutions, and the IT business sector will be established, resulting in multiple socioeconomic benefits. More specifically, this study will impact the following target groups: (1) Preschool children and their families. The results arising from this study will significantly improve the conditions of the children’s stay at preschool institutions in the context of the health and safety of children with food allergies and food intolerance. Furthermore, it is expected that this study will build knowledge about proper nutrition, child growth, and development and allow its ready transfer (via mobile and web applications) to children with nutritional difficulties and their parents or caregivers. This will directly contribute to the implementation of protection and prevention measures minimizing the risk of allergic reactions to food (especially severe reactions) and, moreover, minimizing the risk of health impairment due to conditions resulting from imbalanced nutrition and improper eating habits in children; (2) Preschool institutions. As a result of this study, preschool institutions’ personnel will be additionally trained to take care of children with special dietary requirements and compile and adjust their menus to reduce risks and contribute to the well-being of all children, including those with food allergies or intolerance. Furthermore, the institutions that will use the Allergy registry and implement novel guidelines will have the opportunity to be specially certified as the ones that meet the criteria of proper nutrition. The use of the IT platform developed in this study will enable the adjustment of food menus in real-time according to the number of children, their requirements, and available (seasonal) ingredients, which will contribute to significant financial savings. All of the above will potentiate competitiveness and improve the quality standards of educational institutions; (3) Healthcare providers. The development of the Allergy registry will provide pediatricians, general practitioners, and allergy specialists with relevant information which will ensure adequate care and the highest standards in the clinical management of children with food allergies and intolerance. 

Furthermore, the knowledge gained within this study, the development of the new IT platform, and the creation of the Allergy registry will serve as a basis for future studies in the field of food allergy. The use of the functionality of the Allergy registry by clinicians and clinical researchers will drive additional research focused on the mechanisms underlying food allergy and intolerance, the development of novel early biomarkers for food allergy and the development of novel and personalized treatment and prevention options, involving an even larger number of participants, different populations and other European countries.

Possible limitations of the study may arise from the possible lack of cooperation with parents and personnel of preschool institutions, which will be addressed and minimized by good communication, education, and active follow-up. Potential bias includes the overrepresentation of children with special dietary requirements other than food allergy/intolerance due to their parents’/caregivers’ special interest in participating in the study, which will be mitigated by detailed allergy diagnostics in selected individuals lead by a pediatric allergy specialist with 20+ years of experience in the field. Additionally, children with food allergy/intolerance may be underrepresented in some preschool institutions, places (large cities or rural areas), different regions, etc., due to a lack of know-how and facilities to support special dietary requirements. These differences will be taken into account during recruitment and data collection.

To the best of our knowledge, this kind of IT platform, which interconnects preschool institutions, parents, meal providers, nutritionists, and healthcare providers, has not yet been developed or commercially available, and we are of the opinion that it will contribute to the improvement, optimization, and personalization of the monitoring and management of children with food allergies/intolerance.

The results of this study are planned to be replicated in a larger cohort involving a larger number of preschool institutions and stakeholders. Additionally, they may need to be compared and replicated in other populations (older children- school children).

## 5. Conclusions

This study will give better insight into the prevalence of food allergy and intolerance in preschool children in Croatia and contribute to the optimization of treatment and prevention regimes for these conditions. This study will also provide additional insight into the association of nutrition with child growth and development, as well as potential impairments arising from inadequate nutrition. Most importantly, this study will help build a safe environment for children with food allergy/intolerance and related disorders in preschool institutions, thus significantly improving their and their families’ quality of life. 

## Figures and Tables

**Figure 1 medicina-59-01367-f001:**
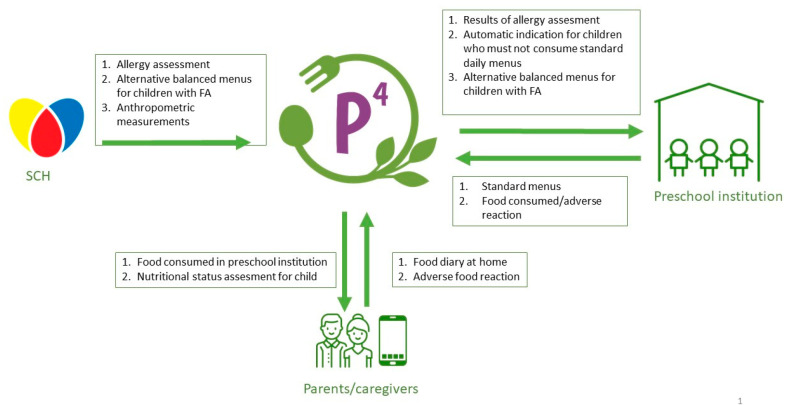
The communication link between different parts and stakeholders in the system via the P4 IT platform: Srebrnjak Children’s Hospital, preschool institutions, and parents/caregivers. The IT platform enables the exchange of data in real time and allows for the possibility of timely adjustment of menus. FA-Food allergy.

**Table 1 medicina-59-01367-t001:** The features used in this study. ENT- ear, nose, and throat; GERD/LPR- Gastroesophageal reflux disease/Laryngopharyngeal reflux; FENO- fractional exhaled nitric oxide; BAT- basophil activation test, BMI-body mass index.

Baseline Demographics	Gender, Age
Subjective clinical data	Personal and family medical history- atopy status, allergic rhinitis (AR), atopic dermatitis (AD), food allergy, and other comorbidities
Objective clinical data	Skin prick and total and specific IgE test results, hematologic and biochemical blood test results, comorbidity status-ENTexamination, pH probing with impedance for the reflux episodes monitoring for diagnostics of GERD/LPR, lung function testing, FENO, BATresults and oral food challenge test results for confirming or excluding a diagnosis of food allergy, treatment used, nutritional status
Anthropometric data	Height, weight, BMI percentile, body composition
Exposure data	Menus and ingredients used in meal preparation in preschool institutions, other food consumed, personal and family dietary habits, food diaries

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
