# Peer review of "Protocol Development of a Personalized Balanced Nutrition Concept for Preschool Children, Primarily Those with Food Allergies, Using an IT Platform"

_medicina, 2023, doi:10.3390/medicina59081367_

Round 1

Reviewer 1 Report

This is a study protocol for the development of an informatic platform which will allow to implement personalized nutrition in preschool children who require a special diet, e.g. infants with food allergies. Authors aim to develop tailored menus in day care facilities, to satisfy nutrient requirements and to adopt healthy eating habits. The informatic system will also establish a communication link between preschool institutions and the Croatian Reference centre for clinical allergy, to allow possible adjustments of the menus in real time.

The rationale of this project is interesting, because the management of food allergies is still difficult and children with food allergies are often at high risk of nutritional inadequacies.). Some aspects however should be clarified; my comments/criticisms as following:

Major comments:

- generally, “preschool” children refer to children aged 3 to 5 years.  An age range 1 to 7 years is reported in the study population: authors should consider editing the term “preschool” or the selected age range

-The study describes a protocol study (see paragraph 2.1 A minimum of 450 pediatric participants will be recruited in a prospective, non-interventional type of clinical study by SCH..) ; thus, it is not clear to the readers why the authors use the past tense in the same paragraph (..Criteria for exclusion “children were assessed by a specialist physician (pediatric allergy specialist). The children were enrolled in the study after ..

Please clarify what the protocol study refers to

- Diagnostic procedures and data collection (paragraph 2.2): will the sample include only children with food allergies, or children randomly selected from preschool institutions? How will food allergy be diagnosed?

I am puzzled to see that all children will be undergo SPT, sIgE, BAT Test, total and allergen-specific immunoglobulin 118 E (sIgE), basophil activation test (BAT), component-resolved diagnostics (CRD) and/or ImmunoCAP Immuno‑Solid phase Allergy Chip (ISAC) tests, blood count and biochemistry assays etc.

The Title is “Protocol development of a personalized balanced nutrition concept for preschool children with food allergy using an IT platform”.

However, we read that “Based on the completed standardized questionnaires and preformed SPT, the target groups of preschool children will be stratified according to their allergy status: a) children with an established diagnosis of food allergy; b) children at risk of developing mild, moderate, or severe allergic reactions based on atopic background; c) children with no diagnosis of food allergy or intolerance; and d) children with a likely diagnosis of food allergy/intolerance based on symptoms ..

Hence, which is the objective of the study? Focusing on “allergic preschool children, as the Title states or anything else?

It is not clear, as well as the inclusion criteria are not clear to the readers

Moreover, in regard to the diagnosis of food allergy It is well known that positivity to SPT, IgEs or other tests alone is not a sufficient basis for making a certain diagnosis of food allergy and starting an avoidance diet. Won’t OFC be performed to confirm diagnosis of food allergy (see point a above)? Authors should better clarify inclusion criteria, since this is a critical point in the development of the platform.

The Title is “Protocol development of a personalized balanced nutrition concept for preschool children with food allergy using an IT platform”, but in Study  population the authors also refer to “food intolerance” (also see Discussion section)

-Discussion: I was a bit confuse about the aim of the study reading that “This study aims to address the issues that significantly impact the quality of life of  children with food allergy and food intolerance and their families as well as potentially cause growth and development impairment for these children

It not seems the quality of the life is a main objective of the study. If so, The Discussion should be partly re-organized and better focused on the main objective of the study, after clarifying  aim , inclusion and exclusion criteria (see above)

In addition, some minor issues should be addressed:

- Define IT platform (at least the first time the term is used)

- Line 90: authors should specify informed “written” consent

- Lines 95 to 106 (inclusion and exclusion criteria): future tense and not past tense should be used for this paragraph

- Line 110: please add more details about how will body composition be assessed

- Term “certain” is recurrent in the paper. Authors should consider avoiding such generic terms

- Table 1: a legend for abbreviations should be used in the table

- line 233: “caregivers” whould be more appropriate than “guardians”

- Authors should implement  the references list with recent literature on the topc of personalized nutrition in food allergy (see and cite Santos MJL, et al. Food Allergy Education and Management in Schools: A Scoping Review on Current Practices and Gaps. Nutrients. 2022 Feb 9;14(4):732. doi: 10.3390/nu14040732; D'Auria E, et al. Personalized Nutrition Approach in Food Allergy: Is It Prime Time Yet? Nutrients. 2019 Feb 9;11(2):359. doi: 10.3390/nu11020359)

In addition, the paper should be revised for English mistakes and typo (e.g. line 111 skincprick, line 131 visitis, line 152 scuh as, line 240 allergiy etc).

/

Reviewer 2 Report

- Please include the table of most common food allergies reported in Croatia.
- Authors planned the minimum sample size (i.e. 450) across 3 distinct geographical regions. However, authors might want to include more details about planned experiments, e.g. sample size from each region, common food allergies from each region, PM2.5 and PM10 conc. associated with zip codes (especially during the wildfire seasons), month (spring and fall), or related factors that might affect total IgE levels.
- For sIGE, which allergens will be used. It seems like authors will perform different allergen-specific tests for different allergic individuals based on the self-reported allergies. More clarity is needed.
- Will there be any assessment of allergen sIgE from multi-food allergies commonly associated with each other, e.g. peanut and cashew allergies?
- Why only participants with acute episodes will be screened for follow up? Overall, different individuals will have different followup time points. For efficient diagnosis and comparative analysis, I recommend long term screening of participants with both tolerant and acute symptoms, since this study will form a basis for future studies (e.g. mechanism of food allergies).
- In the proposed protocol, algorithms can be built to predict the atopic march, based on the reported allergies. Is this prediction a part of the proposed plan?
- Flowchart of the proposed integrated information system is needed for better understanding the data structure, along with the expected number of features in each component.
- More clarity is needed about the possible means of real-time data exchange and its impact on timely adjustment of the menu. How does the proposed information system allow this?
- Line 183: “In case of an observed problem with the child's growth and development insufficient physical activity, the system will allow timely response in the context of modifying the child's dietary habits, menus at preschool institutions and provide guidelines related to improving nutrition and physical activity”. Will this be done in an automated fashion or will be done under manual supervision of the healthcare provider? Please also provide some parameters that will be used to determine the proper child's growth?
- In the introduction section, authors defined three objectives of this research - develop, adapt and implement an information system. It is not clear how the system will adapt and what problems it will solve?
- Another limitation could be the overrepresentation of children with related food allergies belonging to the same geographical regions or zip codes.

- A statistical review of the manuscript is recommended.

 Acceptable

Round 2

Reviewer 2 Report

N/A